# A Panel Study on Lung Function and Bronchial Inflammation among Children Exposed to Ambient SO_2_ from an Oil Refinery

**DOI:** 10.3390/ijerph16061057

**Published:** 2019-03-23

**Authors:** Fabio Barbone, Dolores Catelan, Riccardo Pistelli, Gabriele Accetta, Daniele Grechi, Franca Rusconi, Annibale Biggeri

**Affiliations:** 1Institute for Maternal and Child Health, IRCCS Burlo Garofolo, via dell’Istria 65/1, 34137 Trieste, Italy; 2Department of Statistics, Computer Science, Applications “G. Parenti” University of Florence, Viale Morgagni, 59, 50134 Firenze, Italy; dolores.catelan@unifi.it (D.C.); gabriele.accetta@gmail.com (G.A.); abiggeri@disia.unifi.it (A.B.); 3School of Respiratory Medicine, Sacro Cuore University, Largo Francesco Vito 1, 00168 Rome, Italy; riccardo.pistelli@unicatt.it; 4Epidemiologia e Prevenzione, no profit social enterprise, via Ricciarelli 29, 20148 Milan, Italy; danielegrechi@virgilio.it; 5Unit of Epidemiology, “Anna Meyer” Children’s University Hospital, Viale Pieraccini 24, 50139 Florence, Italy; f.rusconi@meyer.it

**Keywords:** Spirometry, FeNO, acute respiratory effects, air pollution, oil refinery, Sardinia

## Abstract

To determine the acute effects on respiratory function of children exposed to sulphur dioxide (SO_2_), we conducted two population-based longitudinal investigations near a major oil refinery. We enrolled 233 children, age 8–14, in Sarroch (Italy). The first study entailed five monthly spirometric visits (Panel 5). In a subgroup, children positive for history of respiratory symptoms were tested weekly (20 times) with spirometry and fractional exhaled nitric oxide (FeNO) measurement (Panel 20). Baseline questionnaires and daily diaries were recorded. SO_2_, NO_2_, PM_10_ and O_3_ were measured by monitoring stations. Multiple regression models were fitted. Using a multipollutant model, we found that a 10 µg/m^3^ SO_2_ increase at lag0–2 days determined a percent variation (PV) of −3.37 (90% confidence interval, CI: −5.39; −1.30) for forced expiratory volume after one second (FEV1) in Panel 5 and a PV = −3.51 (90% CI: −4.77; −2.23) in Panel 20. We found a strong dose-response relation: 1-h SO_2_ peaks >200 µg/m^3^ at lag2 days = FEV1 PV −2.49. For FeNO, we found a PV = 38.12 (90% CI: 12.88; 69.01) for each 10 µg/m^3^ SO_2_ increase at 8-h time lag and a strong dose-response relation. Exposure to SO_2_ is strongly associated with reduction of lung function and an increase in airway inflammation. This new evidence of harmful effects of SO_2_ peaks should induce regulatory intervention.

## 1. Introduction

Since the publication of the 2010 final rule of the US Environmental Protection Agency (EPA), which conducted an integrated science assessment (ISA) to revise the primary National Ambient Air Quality Standards (NAAQS) for gaseous sulfur dioxide (SO_2_), sufficient, strong evidence has been gathered of a causal relationship between respiratory morbidity and short-term exposure to SO_2_ [1] in children. The epidemiological evidence for respiratory morbidity is consistent and these effects appear to be independent of other pollutants. To evaluate short-term respiratory effects, clinical studies examine durations of exposure of 5–10 min and lags (between exposure and effects) of 10 min–24 h, while epidemiologic studies generally consider 1-h daily maximum or 24-h average concentrations as exposure measures and multiples of single-day, that is same day, lag1 day, lag2 day or 2-day average or 3-day average and so forth, as lags [1,2,3]. In addition, in order to adequately control the effects of peaks (5–10 min), 1-h or 5-min averaging times have been proposed but 1-h averaging time is generally preferred [4]. This consistent evidence comes from animal toxicological, controlled human exposure and epidemiologic studies. Some of the epidemiologic evidence used SO_2_ as exposure and lung function in children as outcome. Forced expiratory volume after one second (FEV1) was the most frequent short-term outcome measured in these investigations [5,6,7,8]. While some studies observed an inverse association between short-term SO_2_ exposure and lung function in children, a number of others did not observe such an association or considered that the results might have been affected by confounding due to copollutants.

There is also limited available evidence indicating that the effect of SO_2_ on respiratory health outcomes may be “generally robust and independent of the effects of gaseous copollutants, including nitrogen dioxide (NO_2_) and ozone (O_3_), as well as particulate copollutants, particularly PM_2.5_ [1]. However, the evidence is insufficient to conclude that short-term exposure to ambient SO_2_ has an independent effect on lung function in children. The correct lag to identify short-term respiratory function decline is still under discussion. In addition, some researchers have used fractional exhaled nitric oxide (FeNO) to investigate the relationship between exposure to air pollutants and airway inflammation [9,10]. In particular, two studies have examined these effects in association with SO_2_, inducing their authors to conclude that the short-term effects on FeNO in children may be associated with 12–24 h lags after acute SO_2_ exposure. Finally, oxidative damage of SO_2_ has been demonstrated in animal studies. As an example, SO_2_ is a systemic oxidative damage agent on various organs of mice according to Meng Z [11].

Considering the global efforts to reduce human exposure to SO_2_ by substituting solid fuels, particularly coal, with other sources of home heating in urban settings, populations who still experience relatively high levels of SO_2_ exposure are often those living near industrial sites, particularly oil refineries, coal-fired power plants, steel plants and others. In the town of Sarroch (province of Cagliari, Italy), which is the site of one of the largest Mediterranean refineries, up to the year 2008 the total annual emission limit of SO_2_ was 12,000 tons, the average values were approximately 8000 tons [12] resulting in documented high population exposure. Since January 2009, a new regulatory emission limit of 6400 tons has been set in the area [13].

To determine the acute effects on the respiratory function of children exposed to SO_2_ from this major oil refinery and petrochemical plant, we conducted a population-based longitudinal panel investigation involving repeated clinical evaluations in conjunction with air quality monitoring. The specific objectives were to identify possible associations between SO_2_ exposure and respiratory outcomes, regardless of copollutants and the relevant short-term lags in FEV1 and FeNO variations.

## 2. Materials and Methods

### 2.1. Setting

The town of Sarroch is located on the Gulf of Cagliari, Sardinia, Italy. Its population is 5234 inhabitants, according to the 2001 Italian Census; its surface is 67.9 km^2^, of which 10% is occupied by the oil refinery. Distance between the refinery and the centre of town, including the school district, is less than 1 km. Specific toxic air emissions of concern in the area include air quality parameters (SO_2_, NO_2_, PM_10_ and O_3_), benzene, poly-aromatic hydrocarbons (PAH) and metals. As a result of the intense employment of the population in the petrochemical industry, socioeconomic indices in Sarroch are higher compared to regional averages [14].

In 2005, according to the local air quality monitoring network, sulphur SO_2_ levels exceeded 126 times the hourly limit of 350 μg/m^3^ set by Italian law [15] and PM_10_ exceeded 15 times the daily limit of 50 μg/m^3^. In the same year, a survey assessed the respiratory health of children living in South West Sardinia [16]. The DRIAS study used a modified version of the ISAAC (International Study of Asthma and Allergies in Childhood) questionnaire and found increased respiratory symptoms, in particular wheezing symptoms, in 6–10 years old children living in Sarroch [17]. Ethics committee approval (CEI-ASL8Cagliari2006) and individual informed consent from the parents were obtained for this project.

All children, aged 8 to 14 years, attending elementary and middle schools (*n* = 233) in Sarroch (Sardinia, Italy) were enrolled in a Panel study with five monthly spirometric visits from January to June 2007 (PANEL 5). At the beginning of the study period, parents were asked to fill in a modified ISAAC questionnaire [18] for individual and family characteristics, occurrence of respiratory symptoms other than asthma and children’s exposure to known or suspected risk factors. Parents’ questionnaires were completed at home. A further self-administered questionnaire, mainly inquiring on personal smoking habits, was completed by adolescents (12–14 years old) at school.

Of the children enrolled in the PANEL 5 study, all those with a history of asthma diagnosis or who reported wheezing symptoms in the previous 12 months, were entered into a nested panel longitudinal study (*n* = 54) (PANEL 20) entailing weekly visits over the same period (January–June 2007), with repeated lung function testing and FeNO measurements. Concurrently, symptoms and therapy were recorded through daily diaries filled in by parents at home. Subjects were examined in the morning, at school. Height and body weight were measured and each subject answered simple questions on symptoms possibly related to recent respiratory infections. Lung function tests were performed according to the American Thoracic Society guidelines [19] with a portable spirometer (Biomedin 2007, Padua, Italy). The following variables were obtained from the best of 3 reproducible forced expiratory manoeuvres: forced vital capacity (FVC), FEV1 and forced expiratory rates at 25–75% of vital capacity (FEF25–75%). FeNO was then measured online by chemiluminescence using the analyser NIOX-COSMED 2007 (Aerocrine, Stockholm, Sweden) [20] and following recommended procedures [21].

### 2.2. Pollutants Assessment

During the study period, hourly concentrations of SO_2_, NO_2_, PM_10_ and O_3_ were measured by two fixed ambient monitoring stations and one mobile station installed by the Environmental Protection Agency Air Monitoring Network of the Sardinia Region following European positioning guidelines [22]. One fixed monitoring station was located approximately 1000 m north of the town centre, the second in a recreational area northeast of the town hall and the mobile station was set up at the centre of the school courtyard. The Environmental Protection Agency Air Monitoring Network of the Sardinia Region is accredited for the measurement of SO_2_, NO_2_, PM_10_ and O_3_, in accordance with European Commission standard methods.

The hourly measurements of pollutants’ concentrations used in the analyses were treated according to the Meta-analysis of Italian studies on the acute effects of air pollution (MISA) protocol [23]. In brief, for each of the three monitors, we calculated 24-h mean, 0–8 am mean, 8 p.m.–8 a.m. mean,1-h maximum, 3-h maximum or 8-h maximum moving average of SO_2_, NO_2_, PM_10_ and O_3_ concentrations.

A day measure was considered missing when 25% of the SO_2_, NO_2_ and PM_10_ hourly data or 25% of the 8-h moving average for O_3_, were missing. For daily missing data we imputed the mean concentration registered on the same day by the other monitors, proportioned to the average value of the concentration of the missing monitor. Statistical analysis was performed on the series of concentrations derived from the mean of the values of the three monitors. Temperature and relative humidity were measured at the two fixed monitoring stations and averaged [24].

### 2.3. Statistical Analysis

Two primary outcomes were studied in both Panel 5 and Panel 20: lung function indices (FEV1 and FEF25–75%) and FeNO. We specified a series of regression models to investigate the relationship between outcomes and air pollutant concentrations at different time lags adjusting for relevant confounders. Panel 5 consisted of 233 clusters (subjects) with a median of five repeated measurements taken at equally spaced time lags of four weeks. Panel 20 consisted of 54 clusters (subjects) with a median of 15 repeated measurements taken at unevenly spaced time lags of about one week.

Models were specified to account for within-cluster correlations based on hourly data. Since the number of clusters was largely greater than the number of repeated measurements within-cluster and the time lag between measurements was long, making an autoregressive structure implausible, we specified a generalized estimating equation model with a sandwich estimator for standard error [25]. These models were adjusted for subject’s height and weight, gender, passive or active smoke exposure, day of the week, parental education, mould or dampness in the house, traffic intensity, recent respiratory infections, history of steroid use for asthma, temperature and relative humidity. We report estimated model coefficients as percent variation (PV) in FEV1 and FEF25–75% for each 10 mg/m^3^ increase of pollutant concentration. The decision to specify a Gamma response with a log link is justified by the need to stabilize the variance and reduce the skewness of FEV1 measurements. The log link also has the advantage of providing a natural interpretation of regression coefficients in terms of percent change. A Gamma model with a log link implies a log transformation of the expected value of the response variable, while log transforming the raw FEV1 values would be equivalent to modelling the expected value of the logarithm. The two choices are almost equivalent when the coefficient of variation (CV) is below 0.70. In our data, CVs by subjects were all below 0.25.

Some FeNO measurement values were below the limit of instrumental detection of 5 ppb (21 measurements over a total of 141: 14.9%). We considered these values as left censored and specified a Tobit regression model (Tobin 1958). We used the natural logarithm of FeNO measurements as response. A robust sandwich estimator of standard error was employed to account for repeated measurements within-subject [25]. These models were adjusted for gender, active and passive smoking, day of the week, parental education, traffic intensity in the area of residence, respiratory infections in the last week and history of prescription of inhaled steroids in the last 12 months, temperature and relative humidity. Estimated model coefficients were reported as percent variation in FeNO concentration in exhaled air for each 10 mg/m^3^ increase of pollutant concentration. The CVs are below 0.70 for 94% of subjects. For the sake of robustness, we preferred a simple linear Tobit model on the log transformed raw FeNO measurements instead of a generalized linear Tobit model (with a Gamma response and a log link). The Tobit model is a mixture model and this could lead to computational problems with non Gaussian data.

The robust sandwich estimator of standard error in GEE and Tobit models are appropriate when the data consist of a large number of small clusters. GEE models are also vulnerable to informative dropouts. The within-cluster correlation was assumed to be exchangeable. These assumptions were met for the Panel 5 study but may have been violated for the Panel 20 study (54 clusters with a median of 15 repeated measurements) [26]. We ran a sensitivity analysis specifying alternative models, as follows: GEE/Tobit jackknife estimates of standard errors, deleting each cluster or each repeated measurement in turn to control the potential bias in standard error estimate when fitting models with a high number of covariates; a mixed model on a log-transformed response that does not rely on the sandwich estimator of standard error; an autoregressive random effect model for Panel studies with unevenly spaced observations on a log-transformed response, which modifies the exchangeability assumption [27]; a linear Gaussian regression model on a log-transformed response that drops any data hierarchy; a series of distributed lag GEE/Tobit models fitted on the data deleting one day of the week in turn, to check the influence of day of the week on lagged effect estimates.

## 3. Results

### 3.1. Descriptive Subject Data

Of the 300 children attending Sarroch primary and secondary schools during the study period, 233 aged older than 7 years were invited to participate in the Panel 5 study, 205 with spirometric measurements after their parents gave written consent. Of these, 54 subjects comprised the Panel 20 group. Participating subject characteristics for both panel studies are displayed in Table 1. Panel 5 and Panel 20 children were similar for age, height, weight and gender. However, Panel 20 children had lower respiratory function indices, higher frequency of respiratory symptoms, a diagnosis of asthma, use of asthma medications and asthma exacerbations. FeNO levels were much higher among Panel 20 children with a doctor’s diagnosis of asthma.

In Table 2, we report the number of lung function tests and FeNO measurements by week and day of the week in children enrolled in the two panel studies. Overall, 909 valid lung function measurements were obtained for Panel 5. For Panel 20 measurements included 747 spirometric and 864 FeNO assessments. No measurements were taken during week 15 (school holiday), thus, only 19 of the 20 planned weeks yielded measurements. Peaks in SO_2_ concentration (≥100 µg/m^3^ hourly SO_2_) occurring 2 days before the spirometric measurement and peaks in SO_2_ (≥ 10 µg/m^3^ 12-h mean SO_2_) occurring the night before the FeNO measurement are marked with one and two asterisks, respectively, in the table.

### 3.2. Descriptive Exposure Data

Table 3 shows descriptive exposure and meteorological data referring to the 39 days of respiratory measurements taken in the Panel 20 study. SO_2_ daily levels show much greater variability as compared with NO_2_, PM_10_ and O_3_. Moreover, for SO_2_ and not for the other pollutants, maximum hourly values present two orders of magnitude greater peaks than daily averages.

### 3.3. Regression Models for Respiratory Function

The association between indicators of respiratory function and average lag0–2 from multipollutant models adjusted for all relevant covariates are shown in Table 4. When the larger group of Panel 5 children was considered, both FEV1 and FEF 25–75% declined with increasing SO_2_ and O_3_ concentrations while NO_2_ and PM10 showed no effects. In particular, a 10 μg/m^3^ SO_2_ increase determined a PV of −3.37 (90% Confidence Interval, CI: −5.39; −1.30) and −6.99 (90% CI: −11.49; −2.27) for FEV1 and FEF 25–75%, respectively. Results from Panel 20 confirmed the association of declining respiratory function with increasing SO_2_ (FEV1: PV = −3.51; 90% CI: −4.77; −2.23; FEF 25–75%: PV = −3.08; 90% CI: −5.94; −0.13). In addition, FEV1 and FEF 25–75% declined at higher O_3_ concentrations. Increasing PM10 was also associated with decreasing FEV1 but not with FEF 25–75%. Contrary to Panel 5 and, paradoxically, in Panel 20, higher NO_2_ levels were associated with higher FEV1 and FEF 25–75%.

To identify the relevant lags for the effects of SO_2_ daily means in Panel 20, 1-h maximum and 3-h maximum moving averages, distributed lags models adjusted for all relevant covariates were considered (Table 5), in relation with FEV1. At lag2 but not at lag0 nor at lag1, consistent effects on FEV1 were seen for SO_2_ with PV −1.43 (90%CI: −1.99; −0.86) for each 10 μg/m^3^ increase in daily mean, PV −0.05 (90%CI: −0.07; −0.02) for each 10 μg/m^3^ increase in 1 h maximum and PV −0.08 (90%CI: −0.11; −0.04) for each 10 μg/m^3^ increase in 3 h maximum.

We found a strong dose-response relation between higher SO_2_ (1h max lag2) and lower FEV1 in the Panel 20 study in a multipollutant, fully adjusted model (Table 6). Compared to the reference category (SO_2_ < 50 μg/m^3^), for one hour SO_2_ peaks greater than 200 μg/m^3^, FEV1 was reduced by PV −2.49 (i.e., 50 mL).

### 3.4. Regression Models for Fractional Exhaled Nitric Oxide

In the Panel 20 study, we also examined the associations between FeNO and 10 μg/m^3^ increases in pollutant concentration, reported as average within the time intervals 0–8 a.m., 8 p.m.–8 a.m. and at lag0, from multipollutant models adjusted for all relevant covariates (Table 7). SO_2_ concentrations were directly associated with FeNO at 0–8 a.m. (PV = 18.57; 90% CI: 5.29; 33.53) and 8 p.m.–8 a.m. (PV = 38.12; 90% CI: 12.88; 69.01) but not at lag0. Instead NO_2_, PM10 and O_3_ were not associated with FeNO.

In the Panel 20 study, we also found a strong direct dose-response relation between SO_2_ (average 8 p.m.–8 a.m.) and FeNO in a multipollutant, fully adjusted model (Table 8). As compared with the first quartile (SO_2_ < 0.1 μg/m^3^), at the fourth quartile (SO_2_ > 2.64 μg/m^3^) the PV of FeNO was 33.18 (90%CI: 14.53; 54.86), that is the concentration of FeNO had increased from 10.83 ppb to 14.43 ppb.

Results from sensitivity analyses specifying alternative models, as described in the Methods section, are included in Appendix A, Table A1 and Appendix B, Table A2 and confirm those presented in this section.

Furthermore, to evaluate possible effect modification by age, many stratified analyses were performed by age group (8–9, 10–15; and 8–9, 10–11, 12–15). However, never a significant interaction was found. Therefore, results stratified by age are not presented.

## 4. Discussion

This investigation evaluated the respiratory effects of SO_2_ exposure on 233 children aged 8–14 years old, selected in 2007 based on residence in a municipality near a major oil refinery. During the 6-month study period, mean SO_2_ in the 24-h was 4.7 μg/m^3^ (sd 4.3), mean SO_2_ 1-h maximum varied depending on day of lag between 45.4 and 81.1 μg/m^3^ and SO_2_ 1-h absolute maximum measured up to 822.7 μg/m^3^. As demonstrated in a recent cross-sectional evaluation [28,29], children living near this oil refinery had decreased lung function and increased markers of bronchial inflammation as compared with a control group of unexposed children living in a rural area. Also the prevalence of asthma symptoms in the previous 12 months was higher in the exposed area. Also wheezing symptoms were increased in 6–10 years old children living in Sarroch [17].

In this longitudinal analysis, that included repeated measures of both exposure and respiratory function, controlling for copollutants and a number of other covariates, FEV1 significantly declined with increasing SO_2_. In Panel 5, a 10 µg/m^3^ SO_2_ increase in the 3-day moving average (lag0–2), which is a measure describing the overall short-term impact of SO_2_ exposure, determined a PV of −3.37% (90% CI: −5.39; −1.30% *p*-Value = 0.008) for FEV1. Similar results were found for FEF 25–75%. FEV1 declined also with increasing O3 concentrations but was not associated with NO_2_ and PM_10_ in Panel 5.

Children enrolled in Panel 20, a subgroup of Panel 5, were examined every week for 20 weeks and lung function testing and FeNO measurements were performed. The PV of FEV1 decline for each 10 µg/m^3^ increase in SO_2_ (average lag0–2), was almost the same as in Panel 5 (Table 4). This is relevant considering that the range of SO_2_ values in Panel 20 was much wider as it included many more days. In Panel 20, after controlling for copollutants, FEV1 declined with increasing O_3_ and PM_10_. Paradoxically, higher NO_2_ concentrations were directly associated with increased FEV1. These results, which may be due to the very low NO_2_ levels in this study–NO_2_ 3-day moving average (lag0–2): 16.9 µg/m^3^, are inconsistent. For instance, in a sensitivity analysis (Appendix A and Appendix B), from multipollutant distributed lag generalized estimating equation models, adjusted for day of the week, parental education, subject height and weight, gender, passive or active smoking, presence of mold or dampness in the house, recent respiratory infections, history of steroid use for asthma, temperature and relative humidity, the PV for every 10 µg/m^3^ NO_2_ increase were +2.34% (*p* = 0.008) lag0–8 h; −4.88% (*p* < 0.001) lag1; −0.83% (*p* = 0.472) lag2–NO_2_ coefficients denoted an inverse association for lag0, a direct one for lag1 and a negligible and not significant one for lag2.

The main objectives of this investigation were to describe FEV1 variations depending on duration of SO_2_ exposure and to identify the specific lag between time of exposure and change in FEV1. Using Panel 20 data, we were able to identify lag2 as the induction period between increases for all durations of SO_2_ measurements (24-h mean, 1-h maximum and 3-h maximum moving average) and FEV1 decline (Table 5). Although for a 10 µg/m^3^ increase in SO_2_ concentration measured over 24-h, the FEV1 PV decline was 20–30-fold greater as compared with 3-h and 1-h peaks, this depended on the proportional variability of exposure within different measurement durations. If PV is considered per unit of standard deviation, the coefficients of FEV1 at lag2 are of the same magnitude, that is, −1.0, −0.9, −0.9, respectively. In clinical terms, SO_2_ peaks of 1-hour, 3-h and 24-h duration induce consistent effects on respiratory function that become evident 48–72 h after exposure.

The results also confirmed a dose-response curve for the effects of SO_2_ peaks as evaluated by 1-h max measurements (Table 6). In fact, the trend was very strong and for 1-h SO_2_ peaks greater than 200 µg/m^3^ the FEV1 was reduced by 50 mL.

The second objective of this investigation was to assess the association between exposure to air pollutants and in particular to SO_2_ and FeNO as a marker of airways inflammation. Accordingly, within the Panel 20 study frame, we measured FeNO at school between 10 a.m. and 12 a.m. and identified an association between higher SO_2_ concentrations and FeNO increase. This association was significant for the lag of SO_2_ exposure between 8 p.m. and 8 a.m. the following day (Table 7). Furthermore, evidence of a clear dose-response relation was found between SO_2_ measured between 8 p.m. and 8 a.m. and FeNO (Table 8). From a multivariate model, increasing quartiles of SO_2_ predicted increasing FeNO values adjusted for individual, environmental and family variables. We did not find any association between FeNO and NO_2_, PM10 or O3, measured considering different induction periods.

Since the magnitude of the effect of SO_2_ found in our study is large (Table 7: for exposure 8 p.m.–8 a.m. PV 38.12%, compared to 3.69% for PM_10_), PM_10_ cannot be a confounder of SO_2_. Further, NO_2_ levels in Sarroch were very low and the absence of effect due to NO_2_ is plausible. In addition, ozone during the night is negligible. Ozone effect can be estimated using daily average or other metrics like 8 h maximum. However, for FeNO daily averages or other metrics related to exposure occurred 12 h before measurements are inappropriate and bias the results toward the null. Notice that the episodes of SO_2_ exposure peaks occurred in a pattern, which made it impossible to register spirometric and FeNO measurements on the relevant lags for the same episode. This was partially due to the fact that the study design considered morning FeNO readings. In a more recent study, conducted in the Milazzo–Valle del Mela high risk areas, afternoon FeNO assessments were conducted and confirmed the present results [30].

The consistent, specific and independent respiratory effects that we found for SO_2_, the pollutant under examination, represent the major strength of this study. At the same time, the absence of an association with NO_2_ and partially PM_10_ and O_3_, should not be extended to higher or more varied exposure concentrations. The characteristics of this study and the low contrast within this 6-month observation period, do not constitute an appropriate setting to test the respiratory effects of NO_2_, PM_10_ and O_3_ exposure in children.

### 4.1. Strengths and Weaknesses

Selection bias was unlikely in this study since 205 over 233 (88%) and 54 over 54 (100%) children were actually followed-up in Panel 5 and Panel 20, respectively. The high compliance rate was achieved because this study was part of a comprehensive research on environment and health sponsored by the local municipality over a 5-year span. However, we recognize that the selection of children to be enrolled in the Panel 20 study was not based on clinical records but on the parents’ responses to the questionnaire on respiratory health. This implied that the Panel 20 population represented children who were perceived by parents as being more affected (Table 1).

We were unable to examine some children during the follow-up due to absence from school (Table 2): for Panel 20 we had 909/1025 total potential measurements (89%) and, for Panel 5, 747/1026 (73%) spirometric and 864/1026 (84%) FeNO measurements. Missingness did not exhibit any systematic pattern, except during the weeks preceding the Easter and St. Ephisius holidays. In addition, there was no correlation between influenza episodes and missing examinations (Panel 5 Spearman’s rank correlation *r* = −0.10 *p* = 0.87 for spirometric examinations; Panel 20 Spearman’s *r* = 0.14 *p* = 0.56 for spirometry and *r* = −0.24 *p* = 0.33 for FeNO).

Confounders were assessed collecting information from the child’s parents through the ISAAC questionnaire, widely used in asthma epidemiology studies. Although some concern about misclassification of the traffic exposure items has been raised [31], it is of little relevance in the present setting because the municipality of Sarroch is very small and the main roads run far from the village (i.e., less than 10% of the parents defined their residence as close to a heavy traffic road). Environmental exposure to tobacco smoke was also measured by a parents’ questionnaire, while active smoking was assessed through a questionnaire administered to adolescents attending the secondary school. Since we did not investigate active smoking among primary school children, we may have underestimated it among children under 11 years. However, active smoking is very unlikely at this age; we recorded 1 active smoker of 1–5 cigarette per day aged 14 years old, two boys and one girl who smoked less than 4 cigarettes per month aged 12 years old. Recent respiratory infections and other conditions that could interfere with spirometric examinations were checked using a questionnaire and by the pneumologist at the time of the visit. Information on drugs administered to the child was obtained from the parents through a daily diary. Parents recorded any medication given during the week and it is unlikely that they missed to record treatments for respiratory disorders. Despite our attempts, we acknowledge that residual confounding might exist in our study due to other unidentified environmental pollutants. Agents produced by refineries that affect air quality and may cause pulmonary morbidity include (but are not restricted to) hydrocarbons, secondary aerosols and so forth [32]. Should any causative agents be associated also with S02 peaks they act as confounders of the association between SO_2_ and our outcomes of interest.

Some misclassification of individual child exposure may have occurred because we measured pollutants at three ambient monitoring stations and not at the child’s home or through other personal monitors. However, all children lived in the small town centre located close to the point source. Possible alternatives to characterize exposure from environmental sources include dispersion modelling from identified sources of pollution, Bayesian kriging based on a grid of passive dosimeters, land-use regression methods and so forth [33]. We chose to base our statistical analysis on the mean of the values of the three monitors recorded during the study period, because all three monitoring stations were within one kilometre from the main refinery stack, the school and all children’ home addresses. Concurrently, the use of information from a large number of passive dosimeters, placed on a dense grid within this area, only allowed for weekly average estimates and therefore Bayesian kriging would have been useless for the evaluation of shorter term effects [34]. The same applied to dispersion modelling which lacked hourly emission information measured at the stack. In addition, hourly dispersion modelling of SO_2_ concentrations require detailed information on meteorology and emissions from high chimneys.

Completeness of air quality data was assessed independently according to a validated protocol [23]. For the 181 days of the study period (December 2006–June 2007), we retrieved pollutant concentrations from three monitors on 70.7% of the days, from two monitors on 28.7% and from one monitor for 1 day only (0.6%). For Panel 5 we used 45 measurements, for Panel 20 we used 109 measurements (Table 2). Missingness was checked for patterns. Moreover, we performed a sensitivity analysis (Appendix A and Appendix B) applying different modelling approaches because, as reported in the literature, a generalized estimating equation approach may not have been robust to informative missingness. Errors in spirometric measurements on these children were minimized by choosing the best out of three measures taken by trained pneumologists, rather than by health technicians. FeNO measurements followed recommended procedures.

A panel study is designed to detect outcome changes over time. The main focus of the study was to evaluate the within-subject short-term variations in respiratory function caused by variations in pollutant exposures, the long term effect of air pollutants being beyond the scope of this work. The wide variation in pollutant concentrations during the panel days is one of the strength of our study: as shown in Table 2, we registered 13 episodes of high SO_2_ concentrations in different weeks.

### 4.2. External Consistency

A recent meta-analysis and additional studies conducted in different populations addressed the relation between SO_2_ and respiratory outcomes, also considering co-pollutants’ effects [35,36,37,38]. Particularly, the meta-analysis [35] showed a significant association with SO_2_ in children. Their approach allowed the simultaneous analysis of all lags considered in different studies, obtaining one pooled measure of association.

An epidemiologic case-crossover study by Smargiassi et al. [39] studied Canadian children 2-4 years of age living in an area of Montreal within 7.5 km from an oil refinery who were hospitalized or visited an Emergency Department, for asthma. Short-term variations in SO_2_ levels among episodes of increased SO_2_ exposures were associated with a higher number of asthma episodes and the associations were more pronounced for same-day (lag0) SO_2_ levels. In the Canadian study, the SO_2_ 24-h mean measured at two monitoring stations were 11.4 and 17.9 mg/m^3^, respectively. The average SO_2_ 1-h maximum ranged between 33.3 and 61.9 mg/m^3^ and the interquartile (IQ) of the SO_2_ 1-h maximum was 30.9–60.1. In summary, the study by Smargiassi et al. [39], although based on the same type of source of SO_2_ (the oil refinery) and on a mean and peak SO_2_ exposure of the same order of magnitude as our Panel 5 study, differs from ours on a number of accounts. These differences included: child age range (2–4 vs. 8–14 year), study population (asthma cases only vs. whole child population), study design (case-crossover vs. longitudinal, panel), exposure assessment (2 fixed monitors near the refinery and dispersion airmod of SO_2_ estimated at child’s residence vs. average of three ambient monitors in the child’s life environment), outcome of interest (hospitalization or Emergency Department visit vs. pulmonary function measures) and measures of association (odds ratio vs. PV). For all these reasons, a comparison between these results is impossible, although both analyses lean toward the same interpretation of a positive association between higher daily SO_2_ mean and peak concentrations and short-term (<3 days) pulmonary effects.

A population-based study evaluated the association of short-term exposure to increased ambient SO_2_ and daily pulmonary function changes among children aged 6–14 years with and without asthma who resided near a coal-fired power plant in Thailand [6]. In that investigation the only SO_2_ measure was the 24-h mean and no peaks were assessed. However, similarly to our study, outcomes of interest included FEV1 and FEF 25–75% and considered PM_10_ as co-pollutant, the study design was longitudinal and two panels were recruited: one with asthma and another without asthma.

A 4-week panel study on asthmatic children aged 9–14 years was carried out in Windsor (Ontario, ON, Canada) [8] to investigate the acute effects of air pollution on pulmonary function and airway oxidative stress and inflammation. Outcomes of interest were, among others, FEV1, FEF 25–75 and FeNO. Data on pollutants were daily averages of SO_2_, NO_2_, PM_2.5_ from two monitoring stations. Even if the concentrations of SO_2_ were lower than in our study (their maximum IQ range was 39.0 mg/m^3^), they found that an IQ-range increase in 3-day SO_2_ average was associated with significant decreases in FEV1 and FEF 25–75, consistently with our results. Liu et al. (2009) did not find an association with FeNO. However, as presented in their Table 4, there was an increase of 5.8% in FeNO for same-day IQ increase of SO_2_, corresponding to an 18.59 μg/m^3^ SO_2_ increase, that is, a 3% increase for each 10 μg/m^3^. This coincides precisely with the 3% increase for lag0 presented in our Table 7. When we considered the 0–8 a.m. or the 8 p.m.–8 a.m. measurements, the FeNO increase for every 10 μg/m^3^ SO_2_ increase became 18.57 and 38.12, respectively.

The Milazzo–Valle del Mela study [30] included a panel study of 50 children with asthma divided into 9 groups followed-up for one week with daily FeNO and FEV1 measurements and personal recording of air-pollutant concentrations. The results showed a 2.4% (90% CI: 1–4%) decrease in FEV1 for a 10 μg/m^3^ increase in daily average SO_2_ concentrations at lag2 and an 8.1% (90% CI: 3–14%) increase in FeNO for a 10 μg/m^3^ increase in daily average SO_2_ concentrations at lag0–1.

Therefore, our results provide evidence of an effect of SO_2_ on FeNO, now a recognized airway inflammation marker in children [40]. As a comment to these external comparisons, we argue that exposure assessment should be based on hourly monitoring of SO_2_, as effects strongly depend on peaks concentrating in a short time lag.

## 5. Conclusions

Short term effects of exposure to SO_2_ is associated with a reduction of lung function and an increase in airway inflammation among children attending primary school in Sarroch, Sardinia, Italy. Strong dose-response relations demonstrated consistently that peaks, rather than higher mean levels, determine effects on both FEV1 decrease and on FeNO increase. In this study, the effects on FEV1 were highest when a 2-day lag was used. For FeNO, the highest variation was within 12 h or less. Short term public health interventions should capitalize on this evidence on peaks, to protect children from respiratory function decline and inflammation during hours of exposure.

Our results on SO_2_ are specific to children and controlled for other pollutants. Further studies should examine whether these conclusions can be extended also to other populations such as the elderly or adults with comorbidity or concurrent exposure to other known respiratory toxicants.

## Figures and Tables

**Table 1 ijerph-16-01057-t001:** Descriptive statistics for children enrolled in the panel studies in Sarroch (Sardinia), Italy (January–June 2007).

Characteristics of Children Enrolled	Panel 5 *n* = 233	Panel 20 *n* = 54
Age (years), mean (sd)	10.65 (1.85)	10.44 (1.69)
Height ^a^ (cm), mean (sd) Δ = 1.55; 2.34	145.60 (11.49)	143.55 (10.51)
Weight ^a^ (kg), mean (sd) Δ = −0.44; 1.22	40.52 (12.84)	40.99 (12.64)
Sex, *n* (%)		
Males	117 (50.21)	31 (57.4)
Females	116 (49.79)	23 (42.6)
Number of spirometric measurements, median (iqr)	5 (1)	15 (3)
% predicted FEV1 ^b^, *n* (%)		
<80	5 (3)	2 (4)
80–90	16 (8)	9 (17)
90–95	31 (16)	12 (22)
95–99	37 (19)	6 (11)
100+	102 (54)	25 (46)
FEV1 (l), mean (sd)	2.33 (0.61)	2.21 (0.60)
FEF 25–75% (L/s), mean (sd)	2.72 (0.85)	2.47 (0.74)
Wheezing, *n* (%)	32 (14)	25 (46)
Asthma, *n* (%)	18 (8)	18 (33)
Nocturnal cough, *n* (%)	39 (17)	34 (63)
Chest tightness, *n* (%)	11 (5)	10 (18)
Asthma medications, *n* (%)	15 (6)	14 (26)
Absence of symptoms, *n* (%)	175 (75)	9 (17)
Asthma exacerbations, *n* (%)	9 (4)	9 (17)
Number of fractional exhaled NO measurements, median (iqr)		15 (2)
Fractional exhaled NO (ppb), mean (sd)		20.29 (22.42)
Asthma negative, *n* = 36, mean (sd)		13.12 (13.85)
Asthma positive, *n* = 18, mean (sd)		34.63 (29.07)
Inhaled steroids, *n* (%)		12 (22)

^a^ June-January period averages and differences. ^b^ Quanjer et al. 1995.

**Table 2 ijerph-16-01057-t002:** Number of spirometric (S) and fractional exhaled nitric oxide (F) measurements by week and day of the week 2 in children enrolled in the Panel 5 and Panel 20 studies in Sarroch (Sardinia), Italy (January–June 2007).

			Weekday													
			Monday		Tuesday		Wednesday		Thursday		Friday		Saturday		Total	
Week	Panel		S	F	S	F	S	F	S	F	S	F	S	F	S	F
**1**	5		23		40		39		30		35		-		167	
***1***		*20*	-	*8*	*14*	*16 ^b^*	*6*	*6*	*7*	*6*	*10*	*13*	-	-	*37*	*49*
***2***		*20*	-	-	-	-	-	-	*16*	*25*	*16*	*19*	-	-	*32*	*44*
***3***		*20*	-	-	-	-	*33*	*38*	-	-	-	-	*5*	*6*	*38*	*44*
***4***		*20*	-	-	*30*	*45*	-	-	-	-	-	-	*4*	*5*	*34*	*50*
**5**	5		20		75		27		28		22		18		190	
***5***		*20*	-	-	*40*	*44*	-	-	-	-	-	-	*7*	*7*	*47*	*51*
***6***		*20*	-	-	*29*	*31*	-	-	-	-	-	-	*11*	*15 ^b^*	*40*	*46*
***7***		*20*	-	-	-	-	*37*	*39*	-	-	-	-	*11*	*11*	*48*	*50*
***8***		*20*	-	-	-	-	-	-	*34 ^a^*	*38*	-	-	*13*	*13*	*47*	*51*
**9**	5		23		79		24		23		21		23		193	
***9***		*20*	-	-	*42*	*42*	-	-	-	-	-	-	*9^a^*	*9*	*51*	*51*
***10***		*20*	-	-	-	-	*36 ^a^*	*38*	-	-	-	-	*13*	*15*	*49*	*53*
***11***		*20*	-	-	-	-	*41*	*46*	*Easter School Holiday*						*41*	*46*
***12***		*20*	*Easter School Holiday*						-	-	*11*	*13 ^b^*	-	-	*11*	*13*
**13**	5		20		42		23		20		46		29		180	
***13***		*20*	-	-	*5*	*6*	-	-	-	-	*20*	*20*	*17*	*19*	*42*	*45*
***14***		*20*	*36*	*40*	-	-	-	-	-	-	-	-	-	-	*36*	*40*
***15***		*20*	*School Holiday*													
***16***		*20*	-	*23*	-	-	-	-	-	-	-	-	*19 ^a^*	*19 ^b^*	*19*	*42*
**17**	5		41		42		26		21		27		22		179	
***17***		*20*	*19*	*24 ^b^*	-	-	-	-	-	-	-	-	*19*	*20*	*38*	*44*
***18***		*20*	-	-	-	-	*23*	*23 ^b^*	-	-	-	-	*24*	*25*	*47*	*48*
***19***		*20*	*25*	*25*	*24 ^a^*	*25*	-	-	-	-	-	-	-	-	*49*	*50*
***20***		*20*	-	-	-	-	*31*	*35 ^b^*	-	-	-	-	*10*	*12*	*41*	*47*
**Total Panel 5**			127		278		139		122		151		92		909	
***Total Panel 20***			*80*	*120*	*184*	*109*	*207*	*225*	*57*	*69*	*57*	*65*	*162*	*176*	*747*	*864*

*^a^* cells refer to days with peaks (hourly SO_2_ ≥ 100 µg/m^3^) occurring 2 days before the spirometric measurement. *^b^* cells refer to days with peaks (SO_2_ ≥ 10 µg/m^3^ 12-h mean) occurring the night before the FeNO measurement.

**Table 3 ijerph-16-01057-t003:** Air pollutants concentration during Panel 20 study days. Sarroch (Sardinia), Italy (January–June 2007).

Exposure (μg/m^3^)	Mean (sd)	Median	Min	Max	IQ Range
SO_2_ 24-h mean	4.7 (4.3)	3.7	0.6	20.7	4.2
SO_2_ 0–8 a.m.	2.9 (4.2)	1.3	0.2	21.9	2.7
SO_2_ 8 p.m.–8 a.m.	2.2 (2.5)	1.1	0.2	13.3	1.8
SO_2_ Lag1	4.4 (4.9)	3.0	0.6	26.1	3.4
SO_2_ Lag2	5.6 (6.6)	3.0	0.6	29.0	5.5
SO_2_ Lag0–2	4.9 (3.8)	3.7	0.7	18.8	2.9
SO_2_ 1-h max lag0	49.6 (60.5)	29.0	1.5	289.5	53.6
SO_2_ 1-h max lag1	45.4 (57.4)	22.5	1.4	254.3	56.5
SO_2_ 1-h max lag2	81.1 (160.8)	30.3	1.6	822.7	45.9
SO_2_ 3-h max moving average lag0	34.7 (50.1)	16.7	1.5	266.1	37.6
SO_2_ 3-h max moving average lag1	30.1 (41.5)	14.8	1.3	218.7	43.0
SO_2_ 3-h max moving average lag2	50.9 (99.9)	19.6	1.4	466.5	39.6
NO_2_ Lag0–2	11.7 (3.6)	12.5	5.7	16.9	6.9
NO_2_4-h mean	12.9 (5.4)	13.7	2.6	23.2	6.9
NO_2_ 1-h max	40.7 (18.4)	40.1	8.0	79.7	23.7
PM_10_ Lag0–2	21.5 (6.2)	22.2	8.6	32.2	7.4
PM_10_ 24-h mean	22.8 (10.6)	21.7	5.4	48.0	16.5
PM_10_ 1-h max	66.1 (42.1)	54.7	19.0	208.1	61.5
O_3_ Lag0–2	62.8 (10.8)	64.3	35.3	81.5	15.5
O_3_ 8-h max moving average	62.9 (11.5)	63.5	38.5	84.0	20.7
Temperature (°C)	14.1 (4.5)	12.8	6.0	24.2	6.4
Relative humidity (%)	74.8 (12.9)	76.5	50.6	96.2	23.0

**Table 4 ijerph-16-01057-t004:** Percent variation (PV) and 90% confidence interval (90% CI) in FEV1 and FEF 25–75% for each 10 μg/m^3^ increase in pollutant concentration. Sarroch (Sardinia), Italy (January–June 2007).

**FEV1**	**Panel 5**			**Panel 20**		
**Average lag0–2**	**PV ^a^**	**90% CI**	***p*-Value**	**PV**	**90% CI**	***p*-Value**
**SO_2_**	−3.37	−5.39; −1.30	0.008	−3.51	−4.77; −2.23	<0.001
**NO_2_**	1.79	−1.18; 4.85	0.325	3.98	2.28; 5.70	<0.001
**PM_10_**	−0.08	−1.82; 1.70	0.943	−0.89	−1.77; −0.01	0.096
**O_3_**	−1.31	−1.71; 0.90	<0.001	−1.02	−1.6; −0.44	0.004
**FEF 25–75%**	**Panel 5**			**Panel 20**		
**Average lag0–2**	**PV**	**90% CI**	***p*-Value**	**PV**	**90% CI**	***p*-Value**
**SO_2_**	−6.99	−11.49; −2.27	0.016	−3.08	−5.94; −0.13	0.086
**NO_2_**	2.18	−4.51; 9.33	0.601	5.20	0.88; 9.70	0.047
**PM_10_**	−0.59	−4.43; 3.40	0.804	0.82	−0.96; −2.64	0.450
**O_3_**	−1.99	−2.83; −1.14	0.000	−1.97	−3.13; −0.80	0.006

^a^ From a multipollutant generalized estimating equation model, adjusted for day of the week, parental education, subject height and weight, gender, passive or active smoking, mould or dampness in the house, recent respiratory infections, history of steroid use for asthma, temperature and relative humidity.

**Table 5 ijerph-16-01057-t005:** Distributed lag models in the Panel 20 study. Percent variation (PV) and 90% confidence interval (90% CI) in FEV1 for each 10 μg/m^3^ increase in SO_2_ concentration. Sarroch (Sardinia), Italy (January–June 2007).

Distributed Lag Model	FEV1		
SO_2_ (μg/m^3^)	PV ^a^	90% CI	*p*-Value
24-H Mean			
Lag0	−0.22	−1.06; 0.63	0.669
Lag1	0.73	−0.02; 1.48	0.109
Lag2	−1.43	−1.99; −0.86	<0.001
1 h max			
Lag0	0.02	−0.03; 0.07	0.511
Lag1	−0.01	−0.07; 0.06	0.873
Lag2	−0.05	−0.07; −0.02	0.002
3 h max moving average			
Lag0	0.02	−0.05; 0.09	0.608
Lag1	−0.01	−0.10; 0.09	0.927
Lag2	−0.08	−0.11; −0.04	0.001

^a^ From single-pollutant generalized estimating equation models, adjusted for day of the week, parental education, subject height and weight, gender, passive or active smoking, mould or dampness in the house, recent respiratory infections, history of steroid use for asthma, temperature and relative humidity.

**Table 6 ijerph-16-01057-t006:** Dose-response relation between SO_2_ (1 h max lag2) and FEV1 in the Panel 20 study: relative percent variation (RPV) and adjusted FEV1. Sarroch (Sardinia), Italy (January–June 2007).

SO_2_ (μg/m^3^)	FEV1			Adjusted ^b^
1 h max lag2	RPV^a^	90% CI	*p*-Value	FEV1 (l)
<50 (reference category)	0			2.16
50–100	0.64	−0.86; 2.15	0.486	2.17
100–200	−1.38	−3.10; 0.37	0.194	2.13
200+	−2.49	−4.27; −0.67	0.025	2.11

^a^ From a multipollutant generalized estimating equation model, adjusted for day of the week, parental education, subject height and weight, gender, passive or active smoking, mould or dampness in the house, recent respiratory infections, history of steroid use for asthma, temperature and relative humidity. We use the term “relative” percent variation (RPV) to indicate the PV referred to SO_2_ < 50 mg/m^3^. ^b^ Adjusted predicted FEV1 from the model above setting all covariates at the sampling average.

**Table 7 ijerph-16-01057-t007:** Multipollutant models in the Panel 20 study. Percent variation (PV) and 90% confidence interval (90% CI) in fractional exhaled nitric oxide (FeNO) for 10 μg/m^3^ increase in pollutant concentration. Sarroch (Sardinia), Italy (January–June 2007).

FeNO		PV	90% CI	*p*-Value
h 0–8 a.m. ^a^				
	SO_2_	18.57	5.29; 33.53	0.019
	NO_2_	−7.81	−17.45; 2.96	0.226
	PM_10_	2.61	−2.44; 7.91	0.402
h 8 p.m.–8 a.m. ^a^				
	SO_2_	38.12	12.88; 69.01	0.009
	NO_2_	−8.74	−19.51; 3.48	0.232
	PM_10_	3.69	−1.33; 8.98	0.230
Lag0 ^a^				
	SO_2_	3.00	−7.32; 14.47	0.646
	NO_2_	−7.33	−18.52; 5.40	0.331
	PM_10_	−1.56	−6.35; 3.48	0.605
	O_3_	−4.72	−9.72; 0.56	0.141

^a^ From separate Tobit multipollutant models adjusted for day of the week, parental education, age and gender, passive or active smoking, mould or dampness in the house, recent respiratory infections, history of steroid use for asthma, temperature and relative humidity.

**Table 8 ijerph-16-01057-t008:** Dose-response relation between SO_2_ (average 8 p.m.-8 a.m.) and fractional exhaled nitric oxide (FeNO) in the Panel 20 study: relative percent variation (RPV) and adjusted FeNO. Sarroch (Sardinia), Italy (January–June 2007).

SO_2_ (μg/m^3^) Average 8 p.m.–8 a.m.				Adjusted ^b^
(quartiles)	RPV ^a^	90% CI	*p*-Value	FeNO (ppb)
<0.1 (reference category)	0			10.83
0.1–1.13	4.63	−5.64; 16.03	0.472	11.33
1.13–2.64	7.55	−6.61; 23.85	0.397	11.65
2.64+	33.18	14.53; 54.86	0.002	14.43

^a^ From a Tobit multipollutant model adjusted for day of the week, parental education, age and gender, passive or active smoking, mould or dampness in the house, recent respiratory infections, history of steroid use for asthma, temperature and relative humidity. We use the term “relative” percent variation (RPV) to indicate the PV referred to SO_2_ < 0.1 mg/m^3^. ^b^ Adjusted predicted FeNO from model above setting all covariates at the sampling average.

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
