# Peer review of "A Panel Study on Lung Function and Bronchial Inflammation among Children Exposed to Ambient SO2 from an Oil Refinery"

_ijerph, 2019, doi:10.3390/ijerph16061057_

Round 1
Reviewer 1 Report
This is an excellent study which demonstrates the pattern of short-term toxicity of SO2 on the lungs of children. It convincingly shows that exposure to SO2 is strongly associated with measurable changes in lung function and measures of airway inflammation.
If the discussion included consideration of the longer term effects of the refinery emissions it would be more interesting.
Author Response
Point-by-point responses to the reviewers' comments.
Reviewer 1
This is an excellent study which demonstrates the pattern of short-term toxicity of SO2 on the lungs of children. It convincingly shows that exposure to SO2 is strongly associated with measurable changes in lung function and measures of airway inflammation.
If the discussion included consideration of the longer term effects of the refinery emissions it would be more interesting.
A R1: We thank the reviewer for her/his comments. Our study design cannot answer questions about long-term effect of exposure to oil refinery emissions.
Indeed by analogy with nitrogen dioxide, continuous exposure can be associated to pulmonary growth impairment and chronic obstructive pulmonary disease in adulthood (Gauderman, WJ et al. The Effect of Air Pollution on Lung Development from 10 to 18 Years of Age N Engl J Med 2004;351:1057-67. And Gauderman WJ, et al. Association of improved air quality with lung development in children. N Engl J Med. 2015 Mar 5;372(10):905-13). See also: A preliminary result from a follow-up study on the children population of South-West Sardinia including the present study one, is consistent with those literature supporting a similar effect for sulfur dioxide exposure (Cecconi L. Effect of air pollution on lung development in children: the DRIAS follow-up study. Master Thesis. University of Turin. 2015).
Nevertheless, since the topic raised by the reviewer is quite distant from the objective of our study, we felt that there was no need to make changes in the manuscript about this point.
Reviewer 2 Report
Review:
This is an interesting investigation demonstrating the respiratory effects of SO2 exposure on 230+ schoolchildren in an area near to a major oil refinery. Mean SO2 levels were significantly enhanced during the 6-month study period, and children living in this area had decreased lung function (as assessed via FEV1 measurements).
There were two panels of children (Panel 20 and Panel 5) in which lung function testing and FeNO measurements were performed. The great advantage of this work is a number of children included, which was quite a large one (233 kids). The experimental group was well defined and described and all experimental tools were used accordingly.
During the investigation, more than 1600 spirometric examinations and more than 800 FeNO measurements were performed.
Major comments:
The authors aimed to characterize the biological effect of SO2 as possible cause of bronchial inflammation as the second objective of this investigation. However, the only investigation made was a correlation of SO2 with FEV1 and FeNO (dose –response relation). To my opinion this is to low evidence to claim that a biological effect was characterized. Was it only a loco-regional effect or also a systemic component of SO2 activity upon respiratory immunity, particularly, when the effect was evident ca 48-72 hours after exposure?
There is no information about magnitude of this effect in different age groups. One could assume, that this effect might be even higher in smaller children. Please adress this issue by stratifing effects into 2-3 age groups.
Did you perform any mechanistical approach to explain these effects (eg. nasopharyngeal aspiration?) If not, what is already known from animal studies ?
While a lot has been known about detrimental effects of environmental pollution on existing asthmatic phenotype, an interesting point is to ask for some data on asthma and or allergi rhinitis morbidity in areas like Sarroch. Is there any data on that ?
How would you explain lack of any association between FeNO and PM10 or O3 ?
The discussion is overall good, however, the authors almost omitted limitation(s) of this study. Please address it. A good standard of writing a discussion would help a lot: 1. principal findings; 2. Methodology; 3. Strong points and limitations of the study; 4. Potential confounders; 5. Comparison with other studies; 6. Conclusions and future research
What really disturbs me in the discussion was lack of some information why these results are submitted for publication more than 10 years (!) after completion of this study.
The real novelty of the findings should be emphasized both in the abstract and in the discussion.
Author Response
Reviewer 2
Q R2.1The authors aimed to characterize the biological effect of SO2 as possible cause of bronchial inflammation as the second objective of this investigation. However, the only investigation made was a correlation of SO2 with FEV1 and FeNO (dose –response relation). To my opinion this is to low evidence to claim that a biological effect was characterized. Was it only a loco-regional effect or also a systemic component of SO2 activity upon respiratory immunity, particularly, when the effect was evident ca 48-72 hours after exposure?
A R2.1: we agree with the referee that our study is focused in time and space. However, there are reports in the literature regarding systemic effect of exposure to sulfur dioxide (Thompson AM, et al. Baseline repeated measures from controlled human exposure studies: associations between ambient air pollution exposure and the systemic inflammatory biomarkers IL-6 and fibrinogen. Environ Health Perspect. 2010 Jan;118(1):120-4.). In a different study on a different children population exposed to sulfur dioxide from an oil refinery, we found evidence of association between FeNO and epigenetic markers of inflammation (Baccarelli A, et al. Nasal Cell DNA Methylation, Inflammation, Lung Function, and Wheezing in Children with Asthma. Epigenomics, February 2012, Vol. 4, No. 1, Pages 91-100.).
In any case, in the manuscript the claim at line 339 was slightly reformulated according to the reviewer’s view.
Q R2.2: There is no information about magnitude of this effect in different age groups. One could assume, that this effect might be even higher in smaller children. Please adress this issue by stratifing effects into 2-3 age groups.
A R2.2: We performed many dozens of stratified analyses by age group. However, never a significant interaction was found.
For example, the age distribution of the Panel 20 dataset is reported in the table below.
PANEL 20:
AGE | Freq. Percent Cum.
------------+-----------------------------------
8 | 101 13.12 13.12
9 | 137 17.79 30.91
10 | 103 13.38 44.29
11 | 229 29.74 74.03
12 | 77 10.00 84.03
13 | 100 12.99 97.01
14 | 20 2.60 99.61
15 | 3 0.39 100.00
------------+-----------------------------------
Total | 770 100.00
As an example, the percent variation for sulfur dioxide lag 0-2 (with 90%CI) are reported below for age group 8-9 and 10-15 – the results on three age groups suffered of lack of convergence for age group 12+ .
PANEL 20
------------------------------------------------------------------
fev1 | PV z P>|z| [90% Conf. Interval]
-------------+----------------------------------------------------
AGE 8-9 | -4.91 2.04 0.041 -9.04 -0.94
------------------------------------------------------------------
------------------------------------------------------------------
fev1 | PV z P>|z| [90% Conf. Interval]
-------------+----------------------------------------------------
AGE 10+ | -4.16 5.01 <0.001 -5.56 -2.77
------------------------------------------------------------------
The Wald-type chi-square test for interaction was 0.66 (1 degree of freedom) p=0.416 showing no interaction with age.
A statement about absence of evidence of interaction by age was added at the end of the Results section line 287
Q R2.3 Did you perform any mechanistical approach to explain these effects (eg. nasopharyngeal aspiration?) If not, what is already known from animal studies ?
A R2.3: We did not perform any mechanistical approach to explain the effects. There are animal studies documenting a systemic effect through oxidative damage of sulfur dioxide. (e.g. Meng Z. Oxidative damage of sulfur dioxide on various organs of mice: sulfur dioxide is a systemic oxidative damage agent. Inhal Toxicol. 2003 Feb;15(2):181-95.)
A statement about animal studies documenting a systemic effect through oxidative damage of sulfur dioxide was added line 61-63 along with the corresponding new reference
Q R2.4 While a lot has been known about detrimental effects of environmental pollution on existing asthmatic phenotype, an interesting point is to ask for some data on asthma and or allergi rhinitis morbidity in areas like Sarroch. Is there any data on that ?
A R2.4: Yes, we published a paper on a cross-sectional study comparing Sarroch children with children living in a not exposed area (Rusconi F, et al (2011). Asthma symptoms, lung function and markers of oxidative stress and inflammation in children exposed to oil refinery pollution. Journal of Asthma 48(1):84-90.). Children living in Sarroch versus children living in the reference area showed an increase in wheezing symptoms (a prevalence of 12% and an adjusted prevalence ratio = 1.70 [90% confidence interval (CI) = 1.01; 2.86]. Allergic rhinitis prevalence was 8% while in the reference area was 7%. A wider survey on 4122 children living in 9 municipalities in South-West of Sardinia using the ISAAC questionnaire for respiratory symptoms in childhood was published in an Italian journal (R Pirastu, C Bellu, G Accetta, A Biggeri, e il gruppo DRIAS. (2014) DRIAS - Disturbi Respiratori nell’Infanzia e Ambiente in Sardegna. Pneumologia Pediatrica; 54: 11-18), confirming the higher prevalence in the Sarroch area.
These two references were already included. The information was included in the text line 298-299
Q R2.5 How would you explain lack of any association between FeNO and PM10 or O3?
A R2.5: The magnitude of the effect on sulfur dioxide found in our study is large (Table 7: for exposure 8pm-8am PV 38.12%, comparing to 3.69% for PM10. Such large effect estimate is robust to unknown confounders while the same is not true for PM10 (the Evalue for SO2 is ten times the Evalue for PM10). On the other hand, ozone during the night is negligible. Ozone effect can be estimated using daily average or other metrics like 8h maximum. However, for FeNO daily averages or other metrics relating to exposure before 12 hours from measurements are inappropriate and bias the results toward the null (SO2 effect was PV 3.00 90% CI -7.32; 14.47).
These considerations were added in the manuscript: line 349-354
Q R2.6: The discussion is overall good, however, the authors almost omitted limitation(s) of this study. Please address it. A good standard of writing a discussion would help a lot: 1. principal findings; 2. Methodology; 3. Strong points and limitations of the study; 4. Potential confounders; 5. Comparison with other studies; 6. Conclusions and future research
A R2.6: We agree with the reviewer and reorganized the way we describe study limitations: (a) the fact that Panel 20 was not defined on clinical records but on parents’ responses was moved to the strengths and weaknesses subsection line 370-373; (b) we added a new reference and a statement on possible residual confounding by some unmeasured refinery pollutants line 396-400
Q R2.7 What really disturbs me in the discussion was lack of some information why these results are submitted for publication more than 10 years (!) after completion of this study.
A R2.7: Indeed. The results were used in negotiating lower emissions in regulatory ministerial authorizations and most effort were put in environmental and health surveillance. After that we suffered from interruption of financial support by the Municipality to the Sarroch Environment and Health Project. The window of political support was over. In any situation in which science is used for policy and when researchers are pressed to address local population needs, our ethical constraints are to honestly advocate and postpone general scientific outputs. We discussed this point in the 2014British Columbia Lung Cancer Air Quality and Health Workshop: “Air Quality and Health Impacts of Energy Resource Extraction, ProWe cessing, and Transportation” March 10, 2014, Vancouver BC.
We felt that there was no need to make changes in the manuscript about this point.
Q R2.8: The real novelty of the findings should be emphasized both in the abstract and in the discussion.
A R2.8: Our study shows strong internal consistency in describing the dose-response relationship and the timing between SO2 exposure and biological effects, while NO2, PM10 and 03 were irrelevant. This was obtained using up to date advanced design and analysis techniques. Main novelty of this study is the demonstration that S02 peaks are clearly identified as harmful for respiratory health in children.
This was already written in the conclusions, however, we added a last sentence to the abstract. Line 30.
Reviewer 3 Report
A well designed and conducted study with good presentation standard. Demonstrates what one might intuitively reason that SO2 concentrations impact on respiratory function and that there is a dose response effect.
I have a few observations rather than criticisms.
Our own observations were on SO2, NOx and PM10 on adults emergency admissions with unspecified diseases. The mortality outcome was clearly related to the underlying Illness Severity, but modified by the level of air pollutant.
The difference for our observations were that all air pollutant parameters, irrespective of determination by SO2, NOx or PM10, had the same effect on the outcome variable (mortality). Logically therefore I would not anticipate that there should be a selective action of one pollutant but not for another (of course this is possible). As the authors say however, the level of NO (low level exposure) might be a factor.
I am also suspicious of a lag effect. When one tests many different scenarios or models (medians, varying lags, max, min) then one has the statistical problem of multiplicity. Difficult to know if one has introduced an artefact or is looking at a chance effect. For our data the best overall parameter was the average value on the day of admission and lags did not impact. However, the authors are measuring function (not death) so a lag of 24 - 48 hr is very likely to be correct with activation of inflammatory pathways.
Perhaps some comments might be appropriate on these points.
Author Response
Reviewer 3
Q R3 I have a few observations rather than criticisms.
Our own observations were on SO2, NOx and PM10 on adults emergency admissions with unspecified diseases. The mortality outcome was clearly related to the underlying Illness Severity, but modified by the level of air pollutant.
The difference for our observations were that all air pollutant parameters, irrespective of determination by SO2, NOx or PM10, had the same effect on the outcome variable (mortality). Logically therefore I would not anticipate that there should be a selective action of one pollutant but not for another (of course this is possible). As the authors say however, the level of NO (low level exposure) might be a factor.
I am also suspicious of a lag effect. When one tests many different scenarios or models (medians, varying lags, max, min) then one has the statistical problem of multiplicity. Difficult to know if one has introduced an artefact or is looking at a chance effect. For our data the best overall parameter was the average value on the day of admission and lags did not impact. However, the authors are measuring function (not death) so a lag of 24 - 48 hr is very likely to be correct with activation of inflammatory pathways.
Perhaps some comments might be appropriate on these points.
A R3 We appreciates the reviewer’s comments. As already replied to R2, our study conducted was obtained using up to date advanced design and analysis techniques. It shows strong internal consistency in describing the dose-response relationship and the timing between SO2 exposure and biological effects, while NO2, PM10 and 03 were irrelevant. Since the magnitude of the effect of SO2 found in our study is large (Table 7: for exposure 8pm-8am PV 38.12%, compared to 3.69% for PM10), PM10 cannot be a confounder of SO2. Further, NO2 levels in Sarroch were very low and the absence of effect due to NO2 is plausible. In addition, ozone during the night is negligible. Ozone effect can be estimated using daily average or other metrics like 8h maximum.
Rather than on statistical significance which is influenced by multiple comparisons, our conclusions are based on internal consistency which is described in the discussion.
To strengthen our discussion changes were made in lines 349-354.
Reviewer 4 Report
In this study the Authors performed a longitudinal panel study to determine the acute effects on the respiratory function of 233 children exposed to SO2 from a major oil refinery and petrochemical plant in Sarroch (Sardinia, Italy).
Even if the topic is very interesting, it's lacking in different points:
1. The Authors should avoid abbreviations in brackets in the abstract.
2. In the abstract it is not clear what "Panel 5 study" and “Panel 20 study” means. The Authors should clarify this point better.
3. In Introduction the Authors should provide more information about what SO2 is, particularly in order to the effects on human health.
4. In Materials and Methods the Authors should clarify how far the oil refinery is from the town center.
5. In Materials and Methods more information should be provided about the portable spirometer and the NIOX analyzer (series number? Production year?).
6. In Results, in order to make more clear the numbers in the tables, all statistically significant values should be highlighted (i.e. underlined or in bold).
7. In Discussion the Aurthors should also consider and discuss the direct and allergic effects of exposure to outdoor pollutants on respiratory health and how it affects the results of this study (i.e. consider: “Occupational Exposure to Urban Air Pollution and Allergic Diseases. Int J Environ Res Public Health. 2015 Oct 16;12(10):12977-87. doi: 10.3390/ijerph121012977”; “Effect of outdoor air pollution on asthma exacerbations in children and adults: Systematic review and multilevel meta-analysis. PLoS One. 2017 Mar 20;12(3):e0174050. doi: 10.1371/journal.pone.0174050. eCollection 2017. Review.”; Exposure to traffic-related air pollution and risk of development of childhood asthma: A systematic review and meta-analysis. Environ Int. 2017 Mar;100:1-31. doi: 10.1016/j.envint.2016.11.012; and Respiratory Health in Waste Collection and Disposal Workers. Int J Environ Res Public Health. 2016 Jun 24;13(7). pii: E631. doi: 10.3390/ijerph13070631)
In view of this, I deem this paper need major revision.
Author Response
Reviewer 4
We thank the reviewer for her/his questions.
Q R4.1 The Authors should avoid abbreviations in brackets in the abstract.
A R4.1 We were puzzled by this request and checked samples of IJERPH recent abstracts. We found that abbreviations in brackets are commonly used and, therefore, we did not make changes about this.
Q R4.2 In the abstract it is not clear what "Panel 5 study" and “Panel 20 study” means. The Authors should clarify this point better.
A R4.2: This part of the abstract was modified. Lines 18-21
Q R4.3 In Introduction the Authors should provide more information about what SO2 is, particularly in order to the effects on human health.
A R4.2: In the Introduction we cite 10 external references, however, now further information was added lines 38-39
Q R4.4 In Materials and Methods the Authors should clarify how far the oil refinery is from the town center.
A R4.4: Distance was less than 1 km. This information was added: lines 81-82
Q R4.5 In Materials and Methods more information should be provided about the portable spirometer and the NIOX analyzer (series number? Production year?).
A R4.4: Portable spirometer Biomedin 2007 NIOX-COSMED 2007 Information was included lines 109 and 112, respectively
Q R4.6 In Results, in order to make more clear the numbers in the tables, all statistically significant values should be highlighted (i.e. underlined or in bold).
A R4.6: Although we are aware that different schools of thought exist on representing the precision of measures of association, we deliberately did not classify the results as statistically significant or not. In epidemiological analysis we refrain from dichotomizing the results and prefer to report point estimates and confidence intervals. (Sterne and Davey-Smith BMJ 2001, Rothman, Greenland, etc)
Q R4.7 In Discussion the Aurthors should also consider and discuss the direct and allergic effects of exposure to outdoor pollutants on respiratory health and how it affects the results of this study (i.e. consider: “Occupational Exposure to Urban Air Pollution and Allergic Diseases. Int J Environ Res Public Health. 2015 Oct 16;12(10):12977-87. doi: 10.3390/ijerph121012977”; “Effect of outdoor air pollution on asthma exacerbations in children and adults: Systematic review and multilevel meta-analysis. PLoS One. 2017 Mar 20;12(3):e0174050. doi: 10.1371/journal.pone.0174050. eCollection 2017. Review.”; Exposure to traffic-related air pollution and risk of development of childhood asthma: A systematic review and meta-analysis. Environ Int. 2017 Mar;100:1-31. doi: 10.1016/j.envint.2016.11.012; and Respiratory Health in Waste Collection and Disposal Workers. Int J Environ Res Public Health. 2016 Jun 24;13(7). pii: E631. doi: 10.3390/ijerph13070631)
A R4.7 We agree with the reviewer. Her/his suggested references were added to the manuscript. The concept was strengthened line 431-435
Round 2
Reviewer 2 Report
Well done now ! Congratulations.
Reviewer 4 Report
The Authors have made all the corrections requested in the article.